YUCCA auxin biosynthetic genes are required for Arabidopsis shade avoidance

Müller-Moulé Patricia 1
Nozue Kazunari 1
Pytlak Melissa L. 1 2
Palmer Christine M. 1 3
Covington Michael F. 1 4
Wallace Andreah D. 1 5
Harmer Stacey L. 1
Maloof Julin N. jnmaloof@ucdavis.edu 1
1 Department of Plant Biology, University of California , Davis , CA , United States
2 PASCO Scientific , Roseville , CA , United State of America
3 Natural Sciences Department, Castleton University , Castleton , VT , United States
4 Amaryllis Nucleics , Berkeley , CA , United States
5 Clontech Laboratories , Mountain View , CA , United States
McCormick Sheila
Electronic publication date: 2016 Oct 13
Publication date: 2016
Volume: 4
Electronic Location ID: e2574
Received 2016 Jul 1; Accepted 2016 Sep 15
Copyright: ©2016 Müller-Moulé et al.
Copyright year: 2016
Copyright holder: Müller-Moulé et al.
License: This is an open access article distributed under the terms of the Creative Commons Attribution License, which permits unrestricted use, distribution, reproduction and adaptation in any medium and for any purpose provided that it is properly attributed. For attribution, the original author(s), title, publication source (PeerJ) and either DOI or URL of the article must be cited.
License URL: https://creativecommons.org/licenses/by/4.0/

Keywords: Auxin, Shade avoidance, Phytochrome, Photomorphogenesis

Funding: National Science Foundation (NSF) DBI-0227103 IOS-0923752 UC Davis United States Department of Agriculture NIFA project CA-D-PLB-7226-H USDA National Research Initiative 2004-35100-14903 NIH GM069418 NSF 0315738 This work was funded by National Science Foundation (NSF) grant DBI-0227103 and IOS-0923752 to Julin N. Maloof and Cynthia Weinig, UC Davis funds to Julin N. Maloof, United States Department of Agriculture NIFA project (http://www.csrees.usda.gov/, grant number, grant # CAD-PLB-7226-H)) to Julin N. Maloof, USDA National Research Initiative grant number 2004-35100-14903 to Michael F. Covington, and NIH grant GM069418 and NSF grant IOB 0315738 to Stacey L. Harmer. The funders had no role in study design, data collection and analysis, decision to publish, or preparation of the manuscript.

==============================
Plants respond to neighbor shade by increasing stem and petiole elongation. Shade, sensed by phytochrome photoreceptors, causes stabilization of PHYTOCHROME INTERACTING FACTOR proteins and subsequent induction of YUCCA auxin biosynthetic genes. To investigate the role of YUCCA genes in phytochrome-mediated elongation, we examined auxin signaling kinetics after an end-of-day far-red (EOD-FR) light treatment, and found that an auxin responsive reporter is rapidly induced within 2 hours of far-red exposure. YUCCA2, 5, 8, and 9 are all induced with similar kinetics suggesting that they could act redundantly to control shade-mediated elongation. To test this hypothesis we constructed a yucca2, 5, 8, 9 quadruple mutant and found that the hypocotyl and petiole EOD-FR and shade avoidance responses are completely disrupted. This work shows that YUCCA auxin biosynthetic genes are essential for detectable shade avoidance and that YUCCA genes are important for petiole shade avoidance.

Introduction

Because plants are dependent on light for photosynthesis they have developed a complex system of photoreceptors and downstream responses enabling them to optimize growth to their light environment (Kami et al., 2010). One critical aspect of plant light responses is neighbor detection and shade avoidance (Casal, 2013; Gommers et al., 2013). Plants detect the presence of neighbors by changes in the light quality: since photosynthetic tissue absorbs more red light (R) than far-red light (FR), foliar shade uniquely lowers the R:FR ratio. Changes in the R:FR ratio are detected by phytochrome photoreceptors that exist in two photoconvertible forms, the red light absorbing form, Pr, and the far-red light absorbing form, Pfr. In high R:FR conditions, such as direct sunlight, type II phytochromes are converted from Pr to Pfr and translocated from the cytoplasm to the nucleus (Yamaguchi et al., 1999). Once in the nucleus phytochrome binds to and triggers the degradation of a family of bHLH transcription factors known as PHYTOCHROME INTERACTING FACTORS (PIFs), thereby inhibiting elongation and other phenotypes associated with foliar shade or darkness (Ni, Tepperman & Quail, 1998; Park et al., 2004).

The PIF proteins were originally identified as phytochrome binding factors but are now known to be regulated not only by light but also to integrate signals from the circadian clock, high temperature, and hormone signaling (Leivar & Monte, 2014). They have partially overlapping roles in regulating multiple aspects of development, including promotion of cell elongation and inhibition of both seed germination and chloroplast maturation.

Auxin has long been thought to play a role in shade avoidance (Morelli & Ruberti, 2002; Tanaka et al., 2002). As predicted by Morelli and Ruberti, phytochromes were shown to regulate auxin transport through the shoot (Salisbury et al., 2007) and shade treatment was demonstrated to alter localization of the PIN3 auxin transporter (Keuskamp et al., 2010). Shade also increases endogenous auxin levels (Kurepin et al., 2007; Tao et al., 2008) and auxin signaling (Bou-Torrent et al., 2014; Carabelli et al., 2007; Hersch et al., 2014). Disruption of auxin synthesis by mutation of the TRYPTOPHAN AMINOTRANSFERASE OF ARABIDOPSIS1 (TAA1) gene reduced both shade-induced increases in auxin and shade avoidance elongation responses (Tao et al., 2008; Won et al., 2011). Treatment of leaves with an end-of-day far-red pulse (EOD-FR) will convert type II phytochromes from Pfr to Pr and has been found to increase stem elongation (Gorton & Briggs, 1980), similar to low R:FR. Also similar to low R:FR, EOD-FR induces many auxin-responsive genes, while disruption of auxin signaling via the big/doc1 mutant prevents EOD-FR promotion of petiole elongation (Kozuka et al., 2010). These studies strongly implicate auxin in growth responses to shade and EOD-FR.

PIF proteins were first suggested to promote increases in auxin production and sensitivity based on microarray and dose–response studies of plants with perturbed PIF4 and PIF5 expression (Nozue, Harmer & Maloof, 2011). More conclusive evidence came when it was shown that PIF4 regulates auxin biosynthesis in response to high temperature by promoting transcription of auxin biosynthesis genes (Franklin et al., 2011). More recently it has been demonstrated that PIF4, 5, and 7 are required for normal shade avoidance and function by promoting transcription of the YUCCA family of auxin biosynthesis genes and potentiating auxin responsiveness (Hersch et al., 2014; Hornitschek et al., 2012; Li et al., 2012; De Wit, Lorrain & Fankhauser, 2014).

The YUCCA family consists of eleven genes encoding flavin monooxygenases that function in tryptophan-dependent auxin biosynthesis (Cheng, Dai & Zhao, 2006; Mashiguchi et al., 2011; Won et al., 2011; Zhao et al., 2001). They are expressed in developmentally interesting spatiotemporal patterns (Cheng, Dai & Zhao, 2006; Cheng, Dai & Zhao, 2007). These genes are partially redundant: single knockouts often have no obvious phenotypes but double and higher-order combinations have defects in many aspects of development (Cheng, Dai & Zhao, 2006; Cheng, Dai & Zhao, 2007).

Although the phytochrome/PIF/YUCCA/auxin connection seems clear, most yucca mutant combinations that have been examined to date (yucca1, 4 or yucca3, 5, 7, 8, 9) only show minimal to moderate shade avoidance phenotypes (Li et al., 2012; Tao et al., 2008; Won et al., 2011). More recently, as part of a large phenotypic profiling experiment we reported that the yucca2, 5, 8, 9, quadruple mutant has a strong shade avoidance phenotype (Nozue et al., 2015). Because of the centrality of YUCCA genes to the current shade avoidance model, here we analyze that mutant strain in more detail, beginning with why we decided to make the yucca2, 5, 8, 9 quadruple in the first place.

To better understand the role of the YUCCA genes in shade avoidance and EOD-FR response we used live imaging of an auxin reporter (eDR5::Luciferase) and found a rapid increase in auxin response following an end-of-day far-red (EOD-FR) pulse. We found that the kinetics of the eDR5 reporter response to EOD-FR were similar to the kinetics of YUCCA2, 5, 8, and 9 upregulation, suggesting that these genes are the critical YUCCAs for response to EOD-FR. We tested this idea by generating a yucca2, 5, 8, 9 quadruple mutant and found that these genes are essential both for upregulation of the auxin reporter and for both EOD-FR and low R-FR shade-induced increases in hypocotyl and petiole elongation. These results conclusively show that the YUCCA genes are required for a normal EOD-FR and shade avoidance response.

Materials & Methods

Plasmids

eDR5::LUC+ is described in Covington & Harmer (2007). The pZP-eDR5::LUC2 plasmid was constructed in two steps. First, the luciferase+ gene in the eDR5::LUC plasmid (Covington & Harmer, 2007) was replaced with the luciferase2 (luc2) gene (from pGL4.10, Promega, Madison, WI) using the HindIII and XbaI sites in the two plasmids. Second, the eDR5::LUC2 cassette was removed from the resulting plasmid using the BamHI and PstI sites and cloned into the BamHI and PstI sites of pPZPXomegaLUC+ (a derivative of pPZP221 (Hajdukiewicz, Svab & Maliga, 1994) that contains the RbcS E9 polyadenylation region). The resulting plasmid confers resistance to spectinomycin in bacteria and gentamycin in plants.

Plant materials and growth conditions

Plant transformations were performed by floral dip as previously described (Clough & Bent, 1998). eDR5::LUC2 transformants were selected on gentamycin-containing growth media. The T-DNA and transposon insertion lines were obtained from the Arabidopsis Biological Resource Center (ABRC), the Cold Spring Harbor Lab (CSHL) or GABI-Kat. Mutant yucca lines and plants carrying YUCCA promoter-GUS constructs were obtained from Yunde Zhao and have been previously described (Chen et al., 2014; Cheng, Dai & Zhao, 2006). Multiple mutant combinations were obtained by repeated crossing and PCR genotyping using described primers (Chen et al., 2014; Cheng, Dai & Zhao, 2006). Homozygous athb-2 mutants were obtained from SALK line_106790 (Alonso et al., 2003; O’Malley & Ecker, 2010). Homozygotes were identified by PCR genotyping using standard techniques and the primers listed in Table 1. A reverse-transcription PCR assay was used to confirm that no wild-type message was made.

Table 1 PCR primers.

Gene	Primer type	Sequence	Final concentration	
AtHB-2	LBb1	GCGTGGACCGCTTGCTGCAACT	500 nM	
AtHB-2	LP	TTGGTTGAAATAAAACGAAAAGTG	500 nM	
AtHB-2	RP	CGTCACTGATTCCTCTTGAGC	500 nM	
AtHB-2	qPCR	ACATGAGCCCACCCACTAC	200 nM	
AtHB-2	qPCR	GAAGAGCGTCAAAAGTCAAGC	200 nM	
PP2a	qPCR	TAACGTGGCCAAAATGATGC	200 nM	
PP2a	qPCR	GTTCTCCACAACCGCTTGGT	200 nM	
YUC2	qPCR	ACCCATGTGGCTAAAGGGAGTGA	900 nM	
YUC2	qPCR	AATCCAAGCTTTGTGAAACCGACTG	300 nM	
YUC3	qPCR	CGTCCCTTCATGGCTTAAGGACAAC	900 nM	
YUC3	qPCR	GACGCACCAAACAATCCTTTTCTCG	50 nM	
YUC5	qPCR	ATGATGTTGATGAAGTGGTTTCCTCTG	300 nM	
YUC5	qPCR	ATCAGCCATGCAAGAATCAGTAGAATC	300 nM	
YUC6	qPCR	GAGACGCTGTGCACGTCCTA	300 nM	
YUC6	qPCR	AGTATCCCCGAGGATGAACC	300 nM	
YUC8	qPCR	ATCAACCCTAAGTTCAACGAGTG	50 nM	
YUC8	qPCR	CTCCCGTAGCCACCACAAG	300 nM	
YUC9	qPCR	TCTCTTGATCTTGCTAACCACAATGC	300 nM	
YUC9	qPCR	CCACTTCATCATCATCACTGAGATTCC	50 nM	

For seedling stage EOD-FR analysis, seeds were surface sterilized with 70% ethanol, 0.1% TritonX-100 for 5 min, stratified for four days at 4 °C, then sown on medium containing 1/2X MS with minimal organics (Sigma M6899) and 0.7% agar (Sigma A1296). Seeds were grown in custom chambers outfitted with Quantum Devices Snaplite LEDs under short-day (8 hour day/16 hour night) conditions with 35 μmol m−2 s−1 “red” (peak wavelength 670 nm, half power spectral bandwidth 655–685 nm) and 5 μmol m−2 s−1 “blue” (peak wavelength 470 nm, half power spectral bandwidth 455–485 nm). EOD-FR treatment consisted of a 30 min, 14 μmol m−2 s−1 “far-red” (peak wavelength 730 nm, half power spectral bandwidth 715–745 nm) pulse given nightly for 1 or 4 nights before measurement. LED chamber temperature was 21 °C.

For seedling stage high and low R:FR analysis, seedlings were grown in the same custom chambers as described above for seedling EOD-FR analysis. Light conditions were continuous illumination with 35 μmol m−2 s−1 “red” (peak wavelength 670 nm, half power spectral bandwidth 655–685 nm) and 5 μmol m−2 s−1 “ blue” (peak wavelength 470 nm, half power spectral bandwidth 455–485 nm). After 24 hours, “far-red” (peak wavelength 730 nm, half power spectral bandwidth 715–745 nm) illumination was added to bring the red-to-far-red ratio (R:FR) to 2. After an additional 48 hours the R:FR ratio in one chamber was lowered to 0.5 and plants were grown for an additional 4 days. The chambers assigned to high and low R:FR were swapped for each trial.

For analysis of juvenile plants under EOD-FR seeds were sown as above but plants were grown under 12/12 or short day (8 hr light:16 hr dark) conditions at 22 °C in a Conviron E7 chamber for approximately 18 days with cool white and incandescent lights (75 μmol m−2 s−1 PAR, R:FR 1.4). Two days prior to the EOD-FR pulse, plants were transferred to the LED chambers using the same light and temperature conditions as for seedlings (short day 35 μmol m−2 s −1 red, 5 μmol m−2 s −1 blue light; 21 °C.) and then pulsed as above.

For analysis of juvenile plants under high and low R:FR, stratified seeds were sown on soil and grown under long days in a Conviron walk-in chamber with cool white bulbs and far-red LEDs (Orbitec) (16 h light/8 h night; 100 μmol m −2 s−1 PAR, R:FR 1.8, 22 °C). Two week old plants were transferred to shelves in the same chamber with increased FR (100 μmol m−2 s−1 PAR, R:FR 0.5) to stimulate the shade avoidance response or kept under high R:FR for ten days. Leaves were scanned and petiole length measured as described (Maloof et al., 2013). Plants for were grown under these same high R:FR conditions but were not transferred to low R:FR.

For N −1-naphthylphthalamic acid (NPA; Chem Service, PS-343, http://www.chemservice.com) treatment of eDR5::LUC juvenile plants, seeds were sown and grown as above. 24 hours and 1 hour prior to EOD-FR treatment each plate of plants was sprayed with 1.5ml of DMSO containing 100 μM NPA or an equivalent volume of DMSO alone. Powdered NPA was dissolved in DMSO and stored at −20 °C.

Quantitative RT-PCR

Columbia and athb-2 seedlings were grown as described above for seedling EOD-FR except that they had 30 min EOD-FR pulses on days 3 through 7 and were harvested on day 7, one hour after the end of the final EOD-FR pulse. RNA was prepared with Plant RNeasy (Qiagen) and cDNA prepared with Superscript II (Invitrogen). Real-time qRT-PCR was performed using an iCycler IQ™ 5 (Bio-Rad) in self-made buffer (final concentration: 40 mM Tris-HCL, pH 8.4, 100 mM KCl, 6 mM MgCl2, 8% glycerol, 20 nM fluorescein, 0.4× SYBR Green I (Molecular Probes), 1× bovine serum albumin (New England Biolabs), and 1.6 mM dNTPs) using primers described in Table 1, 10 ng of RNA-equivalent cDNA and Taq polymerase. Each of five to six independent cDNA preparations was assayed two times for each transcript analyzed. Data presented are normalized to the expression level of the control gene PP2a (At1g13320; Czechowski et al., 2005). Transcript abundance was calculated using the relative expression software tool (REST-MCS; (Pfaffl, Horgan & Dempfle, 2002)).

GUS staining

Columbia, YUCCA5::GUS, YUCCA8::GUS and YUCCA9::GUS seeds were grown as described for juvenile plants above. On day 2 in the LED chamber half of the plants were treated with an EOD-FR pulse. Two hours after the pulse plants were taken for GUS analysis. Plants were harvested in 80% acetone on ice and kept in acetone for 30 min. They were then washed twice with pre-staining solution (100 mM NaPO4, pH 7.0, 0.1% (v/v) Triton X-100, 2 mM potassium ferrocyanide, 2 mM potassium ferricyanide, 1 mM EDTA), after which they were vacuum-infiltrated for 10 min with GUS-infiltration buffer (pre-staining solution + 1 mM X-gluc). Images were taken with a Zeiss Discovery-V12 stereo microscope and AxioCam MRC (Zeiss).

Imaging and analysis

For hypocotyl length measurements, whole seedlings were placed on transparency film and scanned with a flatbed scanner (Microtek ScanMaker 8700, http://www.microtek.com). For luminescence measurements, 24 hours prior to luciferase imaging each plant plate was sprayed with 1.5 ml of 3 mM D-luciferin (Biosynth AG) in 0.1% Triton X-100. Bioluminescence was captured with an XR/Mega-10Z ICCD camera (Stanford Photonics) and Piper Imaging software (Stanford Photonics) or an iKon M-934 CCD camera (Andor) controlled by LabView software (National Instruments). Photo analysis software ImageJ (Rasband, 1997) was used to measure both hypocotyl lengths and bioluminescence. Subsequent data analysis was performed in R (R Core Development Team, 2016) using base packages and the add-on packages ggplot2 (Wickham, 2009), reshape2 (Wickham, 2007), lme4 (Bates et al., 2015), lmerTest (Kuznetsova, Bruun Brockhoff & Haubo bojesen Christensen, 2014), and arm (Gelman & Su, 2014).

Figure 1 EOD-FR induction of eDR5::LUC luminescence.

(A–C) Mean luminescence of 5-day-old seedlings (A), 3-week-old juveniles (B), or 3-week-old juveniles in the presence of NPA (C) moved to darkness (solid black line) or treated with a 30 minute EOD-FR pulse prior to transfer to darkness (dashed red line). Dotted lines indicate SEM. Time 0 indicates the beginning of the EOD-FR treatment. n = 4 − 11 plants for each treatment. Representative plots for one of three independent experiments are shown. (D–F) False-color images of eDR5::LUC plants. Representative DMSO treated plant 40 (D) or 240 (E) minutes after EOD-FR pulse showing increase in petiole luminescence after treatment. (F) NPA treated plants 240 minutes after EOD-FR do not have observable petiole luminescence but show increased luminescence in the leaves and apices. (G) Mean luminescence of 3-week-old juveniles treated with DMSO (compare with (C)). For growth conditions see “seedling stage EOD-FR analysis” (A) and “analysis of juvenile plants under EOD-FR” (B-G) in Materials and Methods.

Data and scripts

The raw data and scripts to recreate plots are available on GitHub: https://github.com/MaloofLab/Mueller-Moule-PeerJ-2016.

Results and Discussion

End-of-day far-red treatment rapidly increases auxin responses

It is clear that changes in auxin biosynthesis and sensitivity are critical to shade avoidance responses (Bou-Torrent et al., 2014; Hornitschek et al., 2012; Li et al., 2012; De Wit, Lorrain & Fankhauser, 2014). To examine phytochrome/auxin pathway interactions in real-time we used an enhanced version of the synthetic auxin responsive promoter DR5 (Ulmasov et al., 1997) to drive the expression of firefly luciferase (LUC; Welsh & Kay, 2005), eDR5::LUC (Covington & Harmer, 2007). We initially used an end-of-day far-red (EOD-FR) pulse that, like low R:FR, will reduce the amount of active type II phytochromes, increases expression of auxin responsive genes (Kozuka et al., 2010), and increases stem elongation (Gorton & Briggs, 1980). Plants treated with EOD-FR displayed a strong increase in eDR5::LUC bioluminescence peaking two to three hours after the treatment, consistent with prior reports on eDR5::GUS (Carabelli et al., 2007). This response is found in both seedling stage (Fig. 1A) and juvenile (Fig. 1B) plants and occurred in cotyledons, hypocotyls, petioles, the shoot apex, and developing leaves (Figs. 1D and 1E).

To investigate the importance of auxin transport in eDR5::LUC activation we examined the effect of the auxin transport inhibitor N-1-naphthylphthalamic acid (NPA) on eDR5::LUC expression. Plants grown on NPA still responded with a peak of luminescence following an EOD-FR treatment (Fig. 1C), but in this case the increased bioluminescence was limited to the apex and young leaves (Fig. 1F). The magnitude of induction was somewhat lower on NPA because of higher basal luminescence, however the peak strongly resembles the response of the control plants without NPA (Fig. 1G) and occurs within a similar time-frame. These results suggest that auxin transport is not required to generate the peak of auxin reporter expression following EOD-FR treatment but that transport is required for increased auxin signaling in the petiole. Alternatively, it is possible that the lack of signal in the EOD-FR, NPA treated petioles is due to increased IAA conjugation that can occur in the presence of NPA.

Figure 2 Shade and EOD-FR induction of YUCCA genes.

(A) Expression levels of YUCCA genes from a published shade-induction microarray experiment (Sessa et al., 2005). (B) mRNA levels in EOD-FR treated wild-type plants. (C) mRNA levels in EOD-FR treated athb-2 mutant plants. For (B and C) plants were treated for five days with EOD-FR, and samples were taken 1 hour after the last EOD-FR treatment. mRNA levels shown are normalized to untreated plants. Results shown are averages of n = 5 − 6 ± SEM. Asterisks mark statistical significance of induction (* p-value ≤ 0.05, ** p-value ≤ 0.005) calculated by the REST-program (Pfaffl, Horgan & Dempfle, 2002).

Figure 3 Histochemical localization of GUS in transgenic Arabidopsis thaliana plants containing the YUCCA2::GUS,YUCCA5::GUS, YUCCA8::GUS or YUCCA9::GUS constructs.

(A–D) Whole plants. (E–H) Hypocotyls and shoot-apical meristems. (I–L) Leaves. (M–P) Roots.

Shade treatment induces expression of four YUCCA auxin biosynthetic genes

Shade treatment is known to lead to increased expression of some YUCCA auxin biosynthetic genes (Hornitschek et al., 2012; Li et al., 2012; Tao et al., 2008), so it seemed possible that the induction of eDR5 could be due to increased YUCCA expression. However, most studies of yucca mutants have not found strong shade avoidance phenotypes. One explanation for the observed weak shade phenotypes might be redundancy within the YUCCA gene family. To determine if this could be the case we asked which YUCCA genes were induced by EOD-FR or shade treatments. We first analyzed a published microarray data set (Sessa et al., 2005) and found that three members of this family, YUCCA5, 8, and 9, were all significantly and rapidly induced by low R:FR (P < 0.002; Fig. 2A), suggesting that they would be interesting targets for further analyses. A fourth member, YUCCA2, was more modestly induced (P < 0.03). All YUCCA genes returned to pre-induction levels after four days, indicating that they are involved in early response to shade conditions. We used quantitative real-time reverse transcription PCR (qRT-PCR) to confirm that YUCCA2, 5, 8, and 9 are induced after a series of EOD-FR treatments. One hour after the last EOD-FR treatment all four genes were significantly induced with mRNA levels up to 10 times higher than in control plants (Fig. 2B), consistent with previous microarray studies (Li et al., 2012; Tao et al., 2008).

YUCCA genes 2, 5, 8, and 9 are expressed in organs responsive to shade-treatment

To determine whether these genes were expressed in tissues relevant to shade avoidance, we examined staining in YUCCA2, 5, 8, or 9 promoter::GUS fusions (Fig. 3). All four genes were expressed in the hypocotyls and leaf veins (Figs. 3E–3L). YUCCA2 was also expressed strongly in the primary root, whereas the other three expressed more weakly in primary roots (Figs. 3M–3P). The YUCCA2 and 5 genes were expressed in the shoot apical meristem (Figs. 3E and 3F) and in very defined locations in the leaf. In the leaf they were highly expressed in the veins, petioles, and hydathodes (Fig. 3G). In the roots YUCCA5 was highly expressed at the branching points between primary and secondary roots (Fig. 3N), similar to reported patterns of eDR5::LUC (Moreno-Risueno et al., 2010) suggesting that it may play a role in defining these patterns. The YUCCA8 and 9 genes were expressed in a more diffuse pattern in the leaves starting from the leaf margins (Figs. 3K and 3L), similar to previously reported patterns of eDR5::GUS and Ptaa1::TAA1::GUS (Tao et al., 2008). They were also expressed in secondary roots (Figs. 3O and 3P) but not in the petioles or the shoot apical meristem. In summary, these genes are expressed in the main organs where shade induction of eDR5::LUC expression is observed: all four are expressed in leaves and YUCCA2 and 5 also in the shoot apex.

Figure 4 YUCCA genes are required for shade avoidance.

(A–C) Hypocotyl (A, C) or petiole (B) measurements of short day grown plants with (dark red) or without (blue) EOD-FR pulses. Means of n = 17 − 137 plants ± SEM are shown. Representative data from one of three experiments is shown. (D) Petiole lengths of plants grown in long day high (red, simulated sun) or low (dark red, simulated shade) R:FR conditions. Means of n = 48 − 116 petioles ± SEM are shown. (E) Induction of eDR5::LUC2 expression in 15 day-old wild type and yucca2, 5, 8, 9 mutants moved from short day (8L:16D) conditions to darkness (blue line) or treated with a 30 minute EOD-FR pulse (dark red line). Shading indicates 95% confidence interval. Time 0 indicates the beginning of the EOD-FR treatment. Fourteen Col and 10 yucca2589 plants were measured. For growth conditions see “seedling stage WID-FR analysis” (A, C) and “analysis of juvenile plants under EOD-FR” (B, E), and “analysis of juvenile plants under high and low R:FR” (D) in Materials and Methods.

Figure 5 Hypocotyl length of additional lines in simulated sun and shade.

Four independent experiments were performed with a total of 35–150 plants per treatment/genotype combination. For growth conditions see “seedling stage high and low R:FR analysis” in Materials and Methods.

Figure 6 Adult wild-type and yucca mutant lines.

The mutant lines did not show severe morphological defects, although some showed reduced fertility. For growth conditions see “analysis of juvenile plants under high and low R:FR” in Materials and Methods. These plants were only grown in high R:FR.

AtHB-2 is not required for YUCCA induction

The HD-zip transcription factor AtHB-2 is strongly induced by shade and affects both shade-avoidance traits and auxin-responsive processes (Carabelli et al., 1993; Carabelli et al., 1996; Morelli & Ruberti, 2002; Steindler et al., 1999). We were therefore curious if athb-2 mutations would affect YUCCA induction. However, we found full induction of YUCCA2, 5, 8, and 9 in athb-2 mutants (Fig. 2C). Although not statistically significant the induction appears higher in athb-2 than in wild type, perhaps hinting at a compensatory feedback loop. AtHB-2 may primarily affect auxin transport, as previously proposed (Morelli & Ruberti, 2002) but is not required for YUCCA expression.

YUCCA genes 2, 5, 8, and 9 are required for EOD-FR and low R:FR stimulation of auxin signaling and cell elongation

To determine the relative importance of YUCCA genes for EOD-FR or shade-mediated increases in auxin signaling and subsequent hypocotyl and petiole elongation, we constructed a quadruple mutant with insertions disrupting YUCCA2, 5, 8, and 9 (yucQd) and compared this to yucca5, 8, 9 (yucT) and yucca3, 5, 7 8, 9 (yucQt) mutant strains. The yucT and yucQt strains behaved similarly, partially reducing hypocotyl and petiole EOD-FR responses (Figs. 4A and 4B), similar to previous studies of yucca1, 4 or yucQt lines (Li et al., 2012; Tao et al., 2008; Won et al., 2011). In contrast, the quadruple mutant line completely disrupted EOD-FR in hypocotyls (Fig. 4C) and low R:FR growth responses in petioles (Fig. 4D). In separate experiments we also compared hypocotyl low R:FR response in the yucQd strain to yucca2, yucca5, yucca8, and yucca9 single mutants, a yucca1, 4 double mutant strain, and a yucca2, 5, 9 triple mutant strain (Fig. 5). In this assay all strains were shade responsive except for yucQd (Fig. 5). Across these different experiments the only consistent non-responder to low R:FR and EOD-FR is the yucQd. The difference between the yucQd mutant and the yucT and yucQt combinations is that the yucQd mutant is the only line missing the function of all four of the EOD-FR / low R:FR inducible YUCCA genes. Therefore, this result shows that YUCCA2, 5, 8, and 9 act additively and together are required for the shade avoidance response. In growing the mutant lines for these studies we did not observe any severe morphological defects, although yucQd had reduced fertility (Fig. 6).

The failure of the yucQd mutant to show a morphological shade avoidance response suggested that induction of eDR5::LUC2 by EOD-FR was likely also diminished. To investigate this possibility, the eDR5::LUC2 construct was transformed into the yucQd strain and wild-type plants. We found that EOD-FR induction of eDR5::LUC2 expression was essentially abolished in the yucQd mutant (juvenile plants; Fig. 4E). Thus, YUCCA2, 5, 8, and 9 are required for increased auxin signaling in response to EOD-FR and shade for the subsequent induction of hypocotyl and petiole elongation.

Conclusions

The phenotypic plasticity exhibited by plants in response to shade from other plants is visually striking and is of agronomic importance. Accumulating evidence has led to a model whereby inactivation of phytochromes in shade allows accumulation of PIF transcription factors that upregulate YUCCA transcription and a concomitant increase in auxin biosynthesis. Given this model it has been something of a conundrum that multiple yucca mutants retain a significant (albeit reduced) shade avoidance response, leaving open the possibility of a parallel, YUCCA-independent pathway. By creating a multiple mutant that removes all of the shade-inducible YUCCA genes we demonstrate that YUCCAs are essential for measurable shade avoidance responses in the hypocotyl and also the petiole.

We thank Yunde Zhao and Youfa Cheng for sharing seed and reagents prior to publication. We thank Judy Callis and John Labavitch for helpful discussions on this project. Some seed stocks were obtained from the ABRC.

Additional Information and Declarations

Competing Interests

Author Contributions

Data Availability

Melissa Pytlak is an employee of PASCO scientific, Roseville, California and Andreah Wallace is an employee of Clontech Laboratories, Mountain View, California

Patricia Müller-Moulé and Julin N. Maloof conceived and designed the experiments, performed the experiments, analyzed the data, wrote the paper, prepared figures and/or tables, reviewed drafts of the paper.

Kazunari Nozue performed the experiments, analyzed the data, prepared figures and/or tables, reviewed drafts of the paper.

Melissa L. Pytlak conceived and designed the experiments, performed the experiments, reviewed drafts of the paper.

Christine M. Palmer and Andreah D. Wallace performed the experiments, reviewed drafts of the paper.

Michael F. Covington and Stacey L. Harmer contributed reagents/materials/analysis tools, reviewed drafts of the paper.

The following information was supplied regarding data availability:

Raw data and code for making figures is at GitHub: https://github.com/MaloofLab/Mueller-Moule-PeerJ-2016.

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
