# Peer review of "YUCCA auxin biosynthetic genes are required for Arabidopsis shade avoidance"

_PeerJ, doi:10.7717/peerj.2574_

## Round 0.1 · original submission · Minor Revisions

Two reviewers are largely satisfied, but the third raises many points that must be addressed. Since these points do not appear to require extra experiments, your paper is accepted pending revision.

Reviewer 1 ·

Basic reporting

The Material and Methods section needs input from a qualified photobiologist for the proper details to include. Some example include: It is improper to use ‘color’ to describe the output of LED lighting. These should be specified in wavelength and band width. The temperature in the chamber needs, in all cases, to be specified for both day and, if different, night (lines 113-118; 119-127). This is critical as the adaptive growth response to heat interacts with phytochrome signaling through PIF4 (Franklin et al. PNAS 2011) and thus could confuse the data interpretation. Given the nature of these experiments, the R:FR ratio needs to be given for each chamber used in this study. Especially egregious is the mix of cool white florescent and incandescent chamber that could have a huge range of ratios depending on the mix employed. “Simulated shade” is not defined in terms of total light and “simulated sun” is not defined either in terms of quantity nor R:FR ratio.
The abbreviation NPA is used twice (line 128 and 130) before it is defined (line 184) and the source is not given (white NPA breaks down to a purple inactive byproduct, so source and storage conditions are both quite important). The interpretation of the data is a bit over-simplified and fails to take into consideration the interrelationship between DR5/GH3 and auxin conjugation processes. For example, NPA has been shown to dramatically increase the level of amide conjugation at the auxin source (Liu et al. PlPhysiol 2012), but the effect of this metabolic change on DR5 expression is largely unknown. This makes the conclusion (line 189-191) dubious at best. Also, comparing the NPA dark value (C, black line) with the control (G, dark line) shows an almost immediate increase in auxin reporter signal with NPA treatment alone but this was not discussed.

Experimental design

Perhaps the most glaring problem is the confusion of the shade avoidance syndrome (SAS) with EOD-FR. These two fundamentally different processes are equated by these authors who design experiments apparently with the assumption that they are interchangeable. They cite for this logic a classic paper (Gorton and Briggs, 1980) that does not even mention SAS at all. The steady state situation with phytochromes during a continuous physiological shade (high FR:R) would by necessity be quite different, relative to phyA/phyB equilibrium, in contrast to a EOD-FR pulse. This suggests the authors are operating with a fundamental misunderstanding of the phytochrome system, and this is more than a bit disconcerting. This difference between these two dissimilar physiological processes is also born out in their own Figure 2, where YUC2 hardly changes in the SAS response (Fig 2A) but increases some 10 fold in EOD-FR, YUC 6 declines in SAS but increases somewhat in EOD-FR, YUC 5 and 9 seems to increase in parallel in SAS but YUC 5 is significantly less than YUC 9 increase in EOD-FR.

Validity of the findings

Line 199 refers to microarray data…the correct term would be ‘data set’ and this paper MUST be cited in Figure 2A to clearly show it is not their own data.
Figure 3. The figure legend should indicate how the plants were grown – was it with EOD-FR, shade, or what? The “(A-D) Whole plants. (E-H) Hypocotyls and shoot-apical meristems. (I-L) Leaves. (M-P) Roots” should actually be changed to labels on the figure itself.
Figure 4 is a bit of a mess, but is critical to this paper. First, if the figure is labeled correctly, then the legend is wrong - Legend says: “(A-C) Hypocotyl (A-B) or petiole (C) measurements” but figure says A&C are hypocotyl, B&D are petiole. This figure would be improved if panel E and D both had data for YUC35678 and YUC 589 to compare to YUC 2589. Finally, growth conditions are not described here or in methods in terms of details - see lines 124-126.
Figure 6. The nomenclature for the multiple mutations has changed here. In all other figures yuc2589 is written without punctuation but here they added a solidus mark between numbers: yuc2/5/8/9

Additional comments

This manuscript addresses important information about light quality regulation of auxin and plant growth. Much of the work is new and adds information, but the manuscript has some notable issues demanding the attention of the authors.

Reviewer 2 ·

Basic reporting

No comments

Experimental design

No comments

Validity of the findings

Authors may consider adding statistical analyses to data in Fig. 4 A-D.

Additional comments

The context for the detailed analysis of yucca2,5,8, and 9 genes and mutant in shade avoidance is well defined. The manuscript is written in a clear and concise style. Data are robust and conclusions are well stated.
Two small edits needed:
-In the Introduction, pR and pFR should be changed to Pr and Pfr.
-YUC2 is missing in Fig. 3 legend.

Reviewer 3 ·

Basic reporting

The text is well written and has a logical flow. However, the figures need some work. In my copy, labels for panels in Figures 1 and 4 are either missing or were converted to boxes.

Experimental design

As indicated by the authors, this work expands on previously reported studies (Nozue et al. 2015 PLoS Genetics). The data presented here nicely fills in the details left out of the previous paper. There are no issues with the experiments.

Validity of the findings

No problems with this aspect of the manuscript.

---

## Round 0.2 · accepted · Accept

Thanks for addressing the reviewer comments.